# The Acquisition of Multidrug-Resistant Bacteria in Patients Admitted to COVID-19 Intensive Care Units: A Monocentric Retrospective Case Control Study

**DOI:** 10.3390/microorganisms8111821

**Published:** 2020-11-19

**Authors:** Elisa G. Bogossian, Fabio S. Taccone, Antonio Izzi, Nicolas Yin, Alessandra Garufi, Stephane Hublet, Hassane Njimi, Amedee Ego, Julie Gorham, Baudouin Byl, Alexandre Brasseur, Maya Hites, Jean-Louis Vincent, Jacques Creteur, David Grimaldi

**Affiliations:** 1Department of Intensive Care, CUB-Hôpital Erasme, Université Libre de Bruxelles, 1070 Brussels, Belgium; elisagobog@gmail.com (E.G.B.); Fabio.Taccone@erasme.ulb.ac.be (F.S.T.); antonioizzi1201@gmail.com (A.I.); garufi.ale@gmail.com (A.G.); hnjimi@ulb.ac.be (H.N.); amedee.ego@erasme.ulb.ac.be (A.E.); Julie.Gorham@erasme.ulb.ac.be (J.G.); alexandre.brasseur@erasme.ulb.ac.be (A.B.); jlvincent@intensive.org (J.-L.V.); jacques.creteur@erasme.ulb.ac.be (J.C.); 2Department of Microbiology, LHUB-ULB, Université Libre de Bruxelles, 1070 Brussels, Belgium; nicolas.yin@erasme.ulb.ac.be; 3Department of Anesthesiology, CUB-Hôpital Erasme, Université Libre de Bruxelles, 1070 Brussels, Belgium; stephane.hublet@erasme.ulb.ac.be; 4Clinique d’Epidémiologie et d’Hygiène Hospitalière, CUB-Hôpital Erasme, Université Libre de Bruxelles, 1070 Brussels, Belgium; Baudouin.Byl@erasme.ulb.ac.be; 5Department of Infectious Disease, CUB-Hôpital Erasme, Université Libre de Bruxelles, 1070 Brussels, Belgium; Maya.Hites@erasme.ulb.ac.be

**Keywords:** antimicrobial resistance, *Enterobacteriaceae*, subarachnoid hemorrhage, critically illness, infection control, viral pandemic, SARS-CoV-2

## Abstract

Whether the risk of multidrug-resistant bacteria (MDRB) acquisition in the intensive care unit (ICU) is modified by the COVID-19 crisis is unknown. In this single center case control study, we measured the rate of MDRB acquisition in patients admitted in COVID-19 ICU and compared it with patients admitted in the same ICU for subarachnoid hemorrhage (controls) matched 1:1 on length of ICU stay and mechanical ventilation. All patients were systematically and repeatedly screened for MDRB carriage. We compared the rate of MDRB acquisition in COVID-19 patients and in control using a competing risk analysis. Of note, although we tried to match COVID-19 patients with septic shock patients, we were unable due to the longer stay of COVID-19 patients. Among 72 patients admitted to the COVID-19 ICUs, 33% acquired 31 MDRB during ICU stay. The incidence density of MDRB acquisition was 30/1000 patient days. Antimicrobial therapy and exposure time were associated with higher rate of MDRB acquisition. Among the 72 SAH patients, 21% acquired MDRB, with an incidence density was 18/1000 patient days. The septic patients had more comorbidities and a greater number of previous hospitalizations than the COVID-19 patients. The incidence density of MDRB acquisition was 30/1000 patient days. The association between COVID-19 and MDRB acquisition (compared to control) risk did not reach statistical significance in the multivariable competing risk analysis (sHR 1.71 (CI 95% 0.93–3.21)). Thus, we conclude that, despite strong physical isolation, acquisition rate of MDRB in ICU patients was at least similar during the COVID-19 first wave compared to previous period.

## 1. Introduction

The respiratory infection COVID-19 caused by severe acute respiratory syndrome coronavirus 2 (SARS-CoV-2) has spread worldwide, being considered pandemic by the World Health Organization since 11 March 2020 [1].

During this pandemic, health care systems worldwide became overloaded and experienced shortages of intensive care unit (ICU) beds [2] and protective personal equipment (PPE) [3]. In the most severely affected countries, ICUs were filled with only COVID-19 patients, and named “COVID-19 ICUs”; new ICUs and hospital beds had to be opened [4], health care professionals were reassigned and started working longer hours. These factors can contribute to a decreased adherence to infection prevention and control measures and, combined with a high antimicrobial selection pressure [5], may facilitate the emergence of antimicrobial resistance (AMR) [6]. 

AMR is a major health problem [7], responsible for high morbi-mortality [8,9] and an elevated economic burden [10]. The ICU is one of the hospital locations where patients have the highest risk of acquiring multidrug-resistant bacteria (MDRB) [11]. 

However, until now, the risk of MDRB acquisition in the ICU during COVID-19 crisis remains unknown; whether this risk of MDRB acquisition is higher in COVID-19 units compared to normal functioning ICU has not been studied. 

The aim of this study is to describe the rate of MDRB acquisition in COVID-19 ICUs compared with the MDRB acquisition in the same department before the COVID-19 crisis. 

## 2. Methods 

### 2.1. Study Design and Setting

We conducted a single center retrospective case–control study of COVID-19 patients admitted to the ICUs of Erasme Hospital, Brussels, Belgium from March to April 2020. This research was carried out following the rules of the Declaration of Helsinki and was approved by Erasme Hospital Ethics Committee under the protocol number P2020/371. Due to the retrospective nature of the study the need for informed written consent was waived.

Under normal circumstances, our ICU operates with 30 beds divided into five ICUs with 6 single rooms each. We perform nasal and rectal surveillance cultures upon ICU admission and twice a week thereafter. Colonization by Methicillin Resistant *Staphylococcus Aureus* (MRSA), Extended Spectrum Beta-Lactamase (BLSE), Carbapenemase-Producing Enterobacterales (CPE) and Vancomycin Resistant *Enterococci* (VRE) is indicated at the room door and requires health care workers to wear gloves and gown before entry. We have a meeting with the antibiotic stewardship team twice a week.

### 2.2. COVID-19 Crisis Management

During COVID-19 crisis, we increased our ICU beds from 30 to 39, by opening 5 beds in the existing ICUs and building another 4-bed ICU in two operating rooms. Only one 6-bed ICU remained available for non-COVID-19 admissions. The remaining 33 beds were reserved for COVID-19 patients with a high risk of invasive mechanical ventilation. Patients with stable clinical condition on CPAP remained in the wards. 

### 2.3. Patient Selection

#### 2.3.1. Cases

All patients admitted to the COVID-19 ICUs of Erasme Hospital between 15/03 and 30/04, regardless of the final etiological diagnosis, were eligible for inclusion; the sole exclusion criterion was an ICU stay <48 h. These patients were considered as cases and are named COVID-19 patients in this manuscript.

#### 2.3.2. Controls

The admission to the non-COVID-19 ICU was scarce during the study period. Therefore, it was not possible to use this population as control. Given the unusual duration of mechanical ventilation in COVID-19 patients [12] we compared them to patients obtained from a preexisting subarachnoid hemorrhage (SAH) institutional database cohort [13]. These patients were admitted to our ICU from January 2016–2019. Severe SAH patients are young with few comorbidities and can have prolonged ICU length of stay (LOS) [14] and duration of mechanical ventilation [15]. Patients were matched 1:1 according to ICU LOS and the use of mechanical ventilation. Patients with ICU LOS less than ten days were matched with controls with a difference of ±2 days, and patients with ICU LOS >10 days were matched with controls with a ±20% difference in the LOS. When several controls could match one COVID-19 patient we selected the control with the closest ICU LOS. We also attempted to build a second control group using septic shock patients admitted to our unit from January 2016 to January 2019 using the same matching criteria described above. However, we were unable to match 12 patients due to the longer ICU LOS of the COVID-19 patients.

### 2.4. Data Collection and Endpoints

Demographic data and severity scores were collected at ICU admission. We defined the length of exposure as the duration between ICU admission and the day of MDRB acquisition, or between ICU admission and ICU discharge in patients without MDRB acquisition. We collected the following data during the exposure time: antimicrobial use, presence of central venous catheter, urinary tract catheter, mechanical ventilation and the occurrence of surgery. The primary endpoint was the rate acquisition of MDRB in the COVID-19 units. 

### 2.5. Microbiology Data

We considered patients MDRB+ when a MDRB was found in any microbiological specimen. Patients that were MDRB+ within 48 h after admission were considered index cases. Patients that acquired MDRB during ICU stay were considered new cases. Patients that did not acquire MDRB during ICU stay were considered MDRB-. We collected the presence of possible cross-transmission. Cross-transmission was suspected if a patient acquired a MDR pathogen with the same antimicrobial susceptibility and resistance mechanism than another patient hospitalized at the same time in the same unit.

In our center, we perform routine surveillance cultures (rectal swab, tracheal aspirate and urinary cultures) on admission and then twice a week throughout the ICU stay. 

For MDRB detection, rectal swabs are streaked onto selective plates as follow: chromID^®^ CARBA SMART agar (bioMérieux, Craponne, France) for the detection of carbapenemase-producing *Enterobacteriaceae*; MacConkey agar containing ceftazidime (bioTRADING, Mijdrecht Netherlands) for the detection of third generation cephalosporin-resistant *Pseudomonas aeruginosa, Klebsiella spp.* and *Enterobacter spp.*; chromID^®^ VRE agar for the detection of Vancomycin-resistant *Enterococcus faecium*. Identification of MDRB is performed using matrix-assisted laser desorption/ionization time-of-flight analysis (MALDI–TOF). 

Antimicrobial resistance was defined according to breaking points recommended by the European Committee on Antimicrobial Susceptibility Testing (EUCAST) [16] using VITEK2 and disk diffusion method. Carbapenemases OXA-48, KPC, NDM, VIM and IMP were detected via Polymerase Chain Reaction (PCR) analysis or Coris Resist-5 O.O.K.N.V. antigenic detection (Coris BioConcept, Isnes Belgium). VanA and VanB genes were detected via PCR analysis. ESBL-producing *Enterobacteriaceae* and ampC de-repression were identified using detection of synergy on disk diffusion test as recommended by EUCAST. For methicillin-resistant *S. aureus* detection, nasopharyngeal swabs were streaked onto ChromID^®^ MRSA selective plates. MDR *Pseudomonas* and *Acinetobacter spp.* were defined as recommended considering antimicrobial resistance phenotype [17].

### 2.6. Statistical Analysis

Categorical variables are reported as count (%), continuous data that were normally distributed as mean ± standard deviation (SD) and skewed data as median (interquartile range). 

Incidence of MDRB acquisition was calculated by dividing the number of new cases by the total number of patients admitted to the COVID-19 ICUs. The incidence rate was calculated by dividing the number of new cases by the total number of patient days in the COVID-19 ICUs. The same calculations were done for the two historical control cohorts and after splitting them into tertiles according to admission periods. We compared MDRB+ and MDRB– patients in the COVID-19 cohort using Student’s *t*-test, Mann–Whitney test, χ^2^ test or Fisher’s exact test, as appropriate. In order to identify factors associated with the acquisition of MDRB, we compared patients admitted to COVID-19 ICUs with the SAH patients admitted to our ICUs in previous years (2016–2019); however, a comparison was not done with septic shock patients as not all COVID-19 patients could be matched. Cumulative incidence function of the acquisition of MDRB was used to describe the probability of MDRB acquisition at a given time. The Gray’s test was used to test for the differences. Univariate and multivariable regression analyses were performed using the Fine and Gray competing risks proportional hazards regression model. Death was considered as a competing risk factor for the development of MDRB. In the multivariable model only variables that had *p*-value less than 0.2 by univariate analysis were considered. 

All tests were two-tailed and a *p*-value < 0.05 was considered as statistically significant. Data were analyzed using IBM^®^ SPSS^®^ Statistics software, version 26 for Macintosh (IBM, Armonk, NY, USA).

## 3. Results

### 3.1. Characteristics of Patients Admitted to COVID-19 ICUs 

We identified 75 patients admitted to COVID-19 units during the studied period. Three patients were excluded (ICU stay <48 h) letting 72 included patients representing 1104 patient days of exposure. COVID-19 was diagnosed in 69 out of the 72 patients (67 confirmed by RT-PCR and two suspected cases due to typical clinical and radiologic features); the remaining three non-documented COVID-19 patients were kept in the analysis. Of note, five patients (7%) were MDRB carriers at ICU admission. 

Fifty-two patients (72%) were treated with mechanical ventilation. The median ICU LOS in survivors was 33 days (16–52). 

### 3.2. Characteristics of MDRB Acquisition 

Among the 72 patients, 24 (33%) acquired 31 MDRB (seven patients acquired more than one MDRB) during their ICU stay. The incidence density of MDRB acquisition was 30/1000 patient days. Figure 1 shows the isolated MDRB that were mostly *Enterobacteriaceae*. No MRSA were isolated. We identified 16/31 (52%) suspected cross-transmission events involving mainly ESBL *K. pneumoniae*, AmpC derepressed *E. aerogenes* and Vancomycin-resistant *E. faecium.* MDRB positive and negative patients had similar demographic characteristics and severity at admission. Exposure time was longer in MDRB+ patients and antimicrobial therapy was more frequently used. The most frequently used antimicrobial treatment was piperacillin/tazobactam, and its use was associated with a non-significantly higher rate of MDRB acquisition (Appendix A). Steroids were the only immunosuppressive drugs used in our cohort. Seven patients were treated before MDRB acquisition, and of these, three acquired MDRB during their ICU stay.

In terms of clinical outcome, ICU and hospital mortality were similar between the two groups. These results are shown in Table 1.

### 3.3. Risk of MDRB Acquisition in COVID-19 Patients Compared to the Control Cohort

To evaluate if the MDRB acquisition risk had changed during COVID-19 pandemic, we compared the COVID-19 patient cohort with a retrospective cohort of SAH patients hospitalized in the same ICU matched on the need for mechanical ventilation and the ICU LOS. Characteristics of both cohorts are shown in Table 2; SAH patients had a shorter duration of mechanical ventilation and received less antimicrobial therapy during exposure time. 

In SAH patients, the proportion of patients with MDRB acquisition was lower (21 vs. 33%, *p* = 0.2) than in COVID-19 patients. The MDRB distribution is shown in Appendix A. MDRB acquisition incidence density in SAH patients was 18/1000 patient days compared to 30/1000 patient days in COVID-19 patients (Figure 2). Of note, we were not able to match all of the COVID-19 patients to the septic shock patients. Indeed, the septic shock patients had a shorter ICU LOS. Moreover, these patients had many more comorbidities (Charlson score: 5 (3–6)), and half of them were hospitalized in the previous six months (Appendix A), rendering a comparison hazardous. Their rate of MDRB acquisition was 31/1000 patient days. The rate of MDRB acquisition fluctuated across years in the two control cohorts but without a clear increasing trend (Appendix A).

To analyze if being admitted in a COVID-19 ICU during the first wave was associated with a different risk of MDRB acquisition compared to normal ICUs in previous years, we gathered COVID-19 and SAH patients and performed a multivariable analysis using a competitive risk model. We observed that being admitted to COVID-19 ICUs was not independently associated with MDRB acquisition (sHR 1.71 (CI 95%: 0.93–3.12), *p* = 0.08) (Table 3). The univariate comparison between MDRB+ and MDRB- patients is shown in the Appendix A. 

## 4. Discussion

In this retrospective cohort of patients admitted to COVID-19 ICUs for suspected or confirmed SARS-CoV-2 infection, the incidence density of MDRB acquisition during ICU-stay was 30/1000 patient days. After adjustment and competitive risk analysis, the risk of MDRB acquisition tended to be higher, although not reaching statistical significance, in COVID-19 ICUs compared to control usual ICU.

The interpretation of such results must take into account the rate of acquisition of MDRB in the control group. Indeed, it is highly dependent on local parameters, and thus may vary greatly across centers and countries [18]. In another Belgium center 29% of patients acquired MDRB a proportion slightly higher but comparable to our control cohort [19].

Our study was not designed to explore whether the risk of MDRB acquisition is related to the COVID-19 per se or to the disruption in hospital functioning during the pandemic crisis. In the hospital setting, dissemination of MDRB happens through either cross-transmission or environmental sources, being favored at the individual level by antimicrobial therapy selection pressure [20]. In the ICU environment, cross transmission ranges from 23%–53% of patients’ contacts [21,22] due to the frequent and complex cares, which facilitates the contamination of health care workers’ hands and, consequently, the dissemination of MDRB [20]. Applying infection prevention and control measures and monitoring their observance are key factors to prevent MDRB spreading [23].

The possibility of dissemination of MDRB during a viral pandemic had been theoretically mentioned [24], but as many previous pandemic occurred before the antimicrobial resistance era, no data was available before the SARS-CoV-2 pandemic, including the H1N1 pandemic. Experts have raised concern about the dissemination of MDRB during the COVID-19 pandemic [6,25,26] and a report indicates an increase in blood stream infection [27]. In line with these concerns, we observed a high rate of MDRB acquisition in COVID-19 patients at least as high as in non-COVID-19 patients. The rate of MDRB acquisition in the COVID-19 ICUs was numerically higher, although not statistically different, than the rate in a control cohort of patients hospitalized in the previous years in the same ICUs. The lack of statistical significance may be due to the limited power of our study. Moreover, one can consider that two factors should have limited the MDRB acquisition rate: (1) COVID-19 patients were admitted to ICUs that had been entirely emptied and cleaned in contrast with normal ICU admission, which occurs in units where MDRB carriers are already present and (2) the physical isolation of COVID-19 patients should have provided an efficient barrier to MDRB cross-transmission. 

The high rate of MDRB acquisition may be explained by several factors related to the pandemic: shortage of PPE [28], work overload of the ICU staff, ICU overcrowding, reinforcement of less experimented staff leading to a decreased in adherence to infection prevention and control measures [6,25,26,29]. Our study was not designed to decipher the respective role of these different mechanisms but we can venture some hypothesis to explain our findings: first, we faced a gown shortage and needed to use the same gown for several patients. Considering the difficulty of undressing PPE, it is possible that gloves were not systematically taken off at that moment. We could not document this phenomenon in our study. Second, due to the surge of patients, we had to adapt our usual single room policy creating a new four-bed ICU with two occupants per room. Staying in our four-bed ICU was not associated with MDRB acquisition but the number of concerned patients was low so that no definitive conclusion can be drawn on this point. Third, regarding prevention and control measures, we were not able to continue our hand hygiene and catheter-dressing audit. 

Beside infection control measures, an increase in the use of antimicrobials is also a well-known factor associated with MDRB acquisition [30]. Patients with COVID-19, as for other viral infections [31], were initially suspected to have a high risk of bacterial co-infection and secondary nosocomial infections [32]. In addition, initial COVID-19 symptoms may promote the initiation of antibiotic therapy even when there is no bacterial infection [25]. In line with early descriptions [5], we observed a large antibiotic use in our study, which was significantly associated with a higher risk of MDRB acquisition in COVID-19 patients even if confounding variables might exist. Whether this risk is conferred solely by the use of broad-spectrum classes or by any antibiotic use remains to be determined. Finally, we did not use immunosuppressive drugs during the first wave. The administration of dexamethasone or the other immunosuppressive drugs under investigation could theoretically further increase the risk of MDRB acquisition. 

Our study has several limitations. First, generalization may be difficult since these results may depend on our center’s characteristics. Second, we selected a control group without acute respiratory failure. However, ARDS patients are usually older and have more comorbidities than COVID-19 patients [33]. Moreover, they are, by definition, mechanically ventilated. We also tried to match the septic shock patients with our cohort of patients admitted to COVID-19 units. However, the septic shock patients had shorter ICU LOS, which precluded a perfect match. Additionally, they had many more comorbidities than the COVID-19 patients and had frequently been hospitalized in the previous six months. By contrast, SAH patients offered the advantage of being a cohort of patients with little comorbidity similarly to COVID 19 patients in our center, as shown by the similar Charlson index. Moreover, taking into account predictable differences between the two cohorts, we matched the SAH patients to the COVID-19 patients according to the presence of mechanical ventilation and ICU LOS. Nevertheless, confounders may still exist given the limited numbers of patients in our analysis. 

## 5. Conclusions

The rate of MDRB acquisition was high in patients admitted in ICU during COVID-19 first wave. After adjustment, the risk of MDRB acquisition when being admitted to COVID-19 units was higher, but did not reach statistical significance, compared to control patients hospitalized for SAH before the pandemic and matched on ICU LOS and the use of mechanical ventilation. Larger multicentric studies will be necessary to assess how the viral pandemic impacted the MDRB one.

## Figures and Tables

**Figure 1 microorganisms-08-01821-f001:**
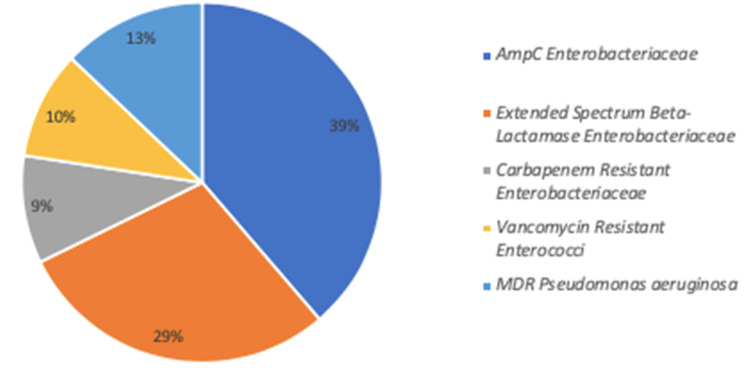
Distribution of MDRB bacteria acquired during COVID-19 ICU stay according to mechanism of resistance. AmpC β lactamase (AmpC) *Enterobacteriaceae* (N = 12): *Enterobacter aerogenes* (N = 6); *E. cloacae* (N = 5); *Escherichia coli* (N = 1). Extended spectrum β-lactamase (ESBL) *Enterobacteriaceae* (N = 9): *Klebsiella pneumoniae* (N = 6); *E. coli* (N = 2); *E. aerogenes* (N = 1). Carbapenem-resistant *Enterobacteriaceae* (CRE) (N = 3): *K. pneumoniae* (N = 2); *E. aerogenes* (N = 1). Vancomycin-resistant *enterococci* (VRE) (N = 3): *Enterococcus faecium* (N = 3); Multidrug-resistant *Pseudomonas aeruginosa* (N = 4).

**Figure 2 microorganisms-08-01821-f002:**
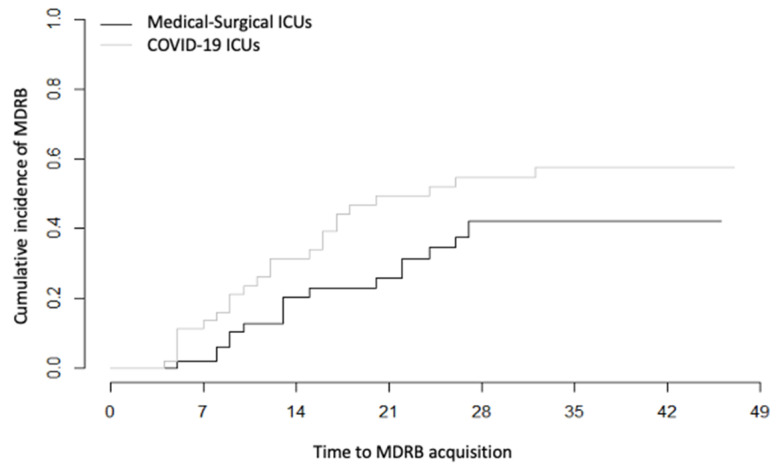
Cumulative incidence of MDRB over time during ICU admission into two types of units (Medical–surgical and COVID-19 units). Comparison of Hazard ratios was calculated using the Fine and Gray method. *p* = 0.14 between groups.

**Table 1 microorganisms-08-01821-t001:** Characteristics of patients that did (MDRB+) or did not (MDRB-) acquire MDRB during their COVID-19 ICU stay.

	All Patients (N = 72)	MDRB- (N = 48)	MDRB+ (N = 24)	*p* Value **
Age; years, mean (±SD)	61 (±14)	62 (±15)	61 (±9)	0.84
Male gender, n (%)	47 (65)	30 (63)	17 (71)	0.60
Charlson comorbidity index, median (IQR)	1(0–3)	1 (0–3)	1 (0–4)	0.77
Hospitalization in the last 6 months, n (%)	11 (15)	9 (19)	2 (8)	0.32
Patients MDRB+ at admission, n (%)	5 (7)	4 (8)	1 (4)	0.66
SAPS 3 score, median (IQR)	56 (47–69)	56 (47–70)	57 (47–65)	0.95
SOFA sore, median (IQR)	6 (3–9)	6 (3–10)	7 (5–9)	0.39
**ICU stay**				
Central venous catheter *, n (%)	68 (94)	45 (93)	23 (96)	0.99
Urinary tract catheter *, n (%)	65 (90)	43 (90)	22 (92)	0.99
Mechanical ventilation *, n (%)	40 (56)	24 (50)	16 (67)	0.22
Length of MV *; days, median (IQR)	3 (0–12)	1 (0–11)	7 (0–15)	0.27
Vasopressor use, n (%)	50 (69)	28 (58)	22 (92)	0.006
Renal replacement therapy, n (%)	17 (24)	10 (21)	7 (29)	0.56
ECMO, n (%)	11 (15)	7 (15)	4 (17)	0.99
Corticosteroid therapy	7 (10)	4 (8)	3 (13)	0.57
Surgery *, n (%)	10 (14)	8 (17)	2 (8)	0.48
Antimicrobial therapy *, n (%)	56 (78)	34 (71)	22 (92)	0.05
Length of Antimicrobial therapy *; days, median (IQR)	5 (2–7)	4 (0–7)	6 (4–7)	0.09
Length of exposure *; days, median (IQR)	9 (4–18)	5 (2–18)	12 (8–18)	0.02
Admission to double bed ICU room *, n (%)	8 (11)	4 (8)	4 (17)	0.43
**Outcome**				
ICU LOS; days, median (IQR)	11 (3–28)	6 (2–18)	28 (17–32)	<0.001
ICU LOS of survivors; days, median (IQR)	7 (2–28)	4 (2–9)	26 (24–28)	0.001
Hospital LOS; days, median (IQR)	24 (12–45)	19 (11–34)	39 (21–61)	0.005
Hospital LOS of survivors; days, median (IQR)	33 (16–52)	20 (11–39)	53 (33–69)	0.002
ICU mortality, n (%)	22 (31)	16 (33)	6 (25)	0.59
Hospital mortality, n (%)	25 (35)	19 (40)	6 (25)	0.30

* variables collected from admission until MDRB acquisition. ** *p*-value was calculated using Qui square test, Fisher exact test, Student’s *t*-test or Mann–Whitney as appropriate. MV: mechanical ventilation; SAPS 3: Simplified Acute Physiology Score III; SOFA: Sequential organ failure assessment; ICU: Intensive Care Unit; ECMO: Extracorporeal membrane oxygenation; IQR: Interquartile range; LOS: length of stay.

**Table 2 microorganisms-08-01821-t002:** Comparison between patients admitted to COVID-19 ICUs and control admitted to medical-surgical ICUs (SAH patients).

	COVID-19 N = 72	Control N = 72	*p*-Value ^#^
Age; years, mean (±SD)	61 (±14)	53 (±16)	<0.001
Male gender, n (%)	47 (65)	32 (44)	0.02
Charlson comorbidity index, median (IQR)	1 (0–3)	1 (0–3)	0.68
Hospitalization in the previous 6 months, n (%)	11 (15)	3 (4)	0.02
Transfer from another hospital **	20 (28)	0 (0)	<0.001
MDRB+ at admission, n (%) ***	5 (7)	4 (6)	0.99
SAPS 3 score, median (IQR)	56 (47–69)	33 (27–39)	<0.001
SOFA sore, median (IQR)	6 (3–9)	5 (1–10)	0.08
Lymphocyte count; G/L, median (IQR) ****	1.07 (0.73–1.74)	1.05 (0.77–1.56)	0.90
**ICU stay**			
Central venous catheter *, n (%)	68 (94)	44 (61)	<0.001
Urinary tract catheter *, n (%)	65 (90)	41 (57)	<0.001
Mechanical ventilation, n(%)	52 (72)	52 (72)	1.0
Length of MV *; days, median (IQR)	3 (0–12)	1 (0–10)	0.55
Vasopressor, n (%)	50 (69)	44 (61)	0.38
Renal replacement therapy, n (%)	17 (24)	1 (1)	<0.001
Antimicrobial therapy *, n (%)	56 (78)	37 (51)	0.002
Length of Antimicrobial therapy * days, median (IQR)	5 (2–7)	1 (0–5)	<0.001
Length of exposure; days, median (IQR)	9 (4–18)	10 (4–22)	0.67
**Outcome**			
MDRB acquisition during ICU stay, n (%)	24 (33)	16 (22)	0.19
ICU LOS; days, median (IQR)	11 (3–28)	11 (3–26)	0.86
ICU LOS of survivors; days, median (IQR)	7 (2–28)	10 (3–27)	0.98
ICU mortality, n (%)	22 (31)	16 (22)	0.26

* variables collected from admission until MDRB acquisition. ** defined as hospital stay >48 h before ICU admission in our center. *** index cases. **** on admission. # *p*-values were calculated using Qui square or Fisher test, Student’s *t*-test or Mann–Whitney as appropriate. SAPS 3: Simplified Acute Physiology Score III; SOFA: Sequential organ failure assessment; ICU: Intensive Care Unit; ECMO: Extracorporeal membrane oxygenation; IQR: Interquartile range; MDRB: multidrug-resistant bacteria; MV: mechanical ventilation.

**Table 3 microorganisms-08-01821-t003:** Uni and multivariable competing risk analyses for factors related to MDRB acquisition.

	Univariable sHR (CI 95%)	Multivariable sHR (CI 95%)
SOFA score	0.97 (0.89–1.05)	
Mechanical ventilation	0.58 (0.30–1.11)	0.61 (0.3–1.26)
Central venous catheter	0.54 (0.26–1.13)	0.66 (0.27–1.62)
Urinary tract catheter	0.98 (0.41–2.33)	
Antimicrobial therapy	1.32 (0.48–3.61)	
Admission to COVID-19 ICUs	1.62 (0.88–2.99)	1.71 (0.93–3.12)

All variables were considered before MDRB acquisition. COVID-19 ICUs: coronavirus disease 19 intensive care units; sHR: sub-distribution hazard ratio. sHR were obtained by the Fine and Gray method for competing risk analysis.

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
