# Peer review of "The Acquisition of Multidrug-Resistant Bacteria in Patients Admitted to COVID-19 Intensive Care Units: A Monocentric Retrospective Case Control Study"

_microorganisms, 2020, doi:10.3390/microorganisms8111821_

Round 1

Reviewer 1 Report

The monocentric retrospective case-control study “The acquisition of Multi-drug resistant bacteria in patients admitted to COVID-19 intensive care units: a monocentric retrospective case control study”, written by Bogossian et al. represents an element of novelty in literature related to Covid-19 crisis. English language and style are fine, all study sections from abstract to conclusions are treated in detail and adequately described, and there is consistency and conformity throughout the article. Introduction and methods are well depicted and referenced, results are clearly presented. Despite the limits intrinsically associated to an observational study, the accuracy in patients’ selection, data collection and statistical analysis make this study believable and interesting. Primary endpoint is well defined.

Noteworthy, authors punctually recognize the limitations of their study: first of all, control group not suffering from acute respiratory failure but, as they explained, cases and controls were matched for LOS and the presence of mechanical ventilation, and had similar Charlson index. Second, they underline the difficulty to generalize their results due to geographic variability of antimicrobial resistance.

Some considerations:

  1. To make this paper more searchable and ensure more citations I suggest to change the keywords, including terms such as “Covid-19” or “viral pandemic”
  2. Inclusion criteria: the study was not designed to explore whether the risk of MDRB acquisition is related to the COVID-19 per se but, specifying the proportion of suspected rather than affected form SARS-Cov-2 patients could help match with the need of antimicrobial use in this category of patients and consequently, help understanding the correlation with MDRB acquisition.
  3. Comparison between patients admitted to COVID 19 ICUs and control admitted to medical-surgical ICUs: as available in Table 2 some variables such as age, lenght of antimicrobial therapy, transfer from another hospital and SAP 3 score are significantly different between cases and controls (p<0.001, not considered in multivariable competing risk analyses for factors related to MDRB acquisition). May these discrepancies influence statistical significance in MDRB acquisition between the two groups?
  4. About large antibiotic use during viral pandemic: authors observed a large antibiotic use in their study, which was significantly associated with a higher risk MDRB acquisition in COVID-19 patients. In our experience, matured in COVID-19 Infectious Disease Unit during pandemic, antibiotics were generally prompted discontinued at the moment of patients transfer to our Unit from the Emergency Room. Currently there is a lack of data about large spectrum antimicrobial use in COVID 19 ICUs and medical wards during first wave pandemic, thus representing a limitation of this study and an area of further research.

    In conclusion, this monocentric retrospective case control study leads the way to further research on the acquisition of multidrug-resistant bacteria in patients admitted to the wards during viral pandemics.

Author Response

Comments and Suggestions for Authors

The monocentric retrospective case-control study “The acquisition of Multi-drug resistant bacteria in patients admitted to COVID-19 intensive care units: a monocentric retrospective case control study”, written by Bogossian et al. represents an element of novelty in literature related to Covid-19 crisis. English language and style are fine, all study sections from abstract to conclusions are treated in detail and adequately described, and there is consistency and conformity throughout the article. Introduction and methods are well depicted and referenced, results are clearly presented. Despite the limits intrinsically associated to an observational study, the accuracy in patients’ selection, data collection and statistical analysis make this study believable and interesting. Primary endpoint is well defined.

We thank the reviewer for these comments.

Noteworthy, authors punctually recognize the limitations of their study: first of all, control group not suffering from acute respiratory failure but, as they explained, cases and controls were matched for LOS and the presence of mechanical ventilation, and had similar Charlson index. Second, they underline the difficulty to generalize their results due to geographic variability of antimicrobial resistance.

Some considerations:

  1. To make this paper more searchable and ensure more citations I suggest to change the keywords, including terms such as “Covid-19” or “viral pandemic”

We thank the reviewer for this suggestion. We have modified the key words to include “viral pandemic” and “SARS-CoV-2”.  “Covid-19”is already in the title and will be caught by keywords research. We expect that these modifications will allow the manuscript to be selected by search tools in several databases.

2.Inclusion criteria: the study was not designed to explore whether the risk of MDRB acquisition is related to the COVID-19 per se but, specifying the proportion of suspected rather than affected form SARS-Cov-2 patients could help match with the need of antimicrobial use in this category of patients and consequently, help understanding the correlation with MDRB acquisition.

The reviewer’s comment gives us the opportunity to clarify the studied population. Among the 72 patients that fulfilled our inclusion criteria 67 patients (93%) had COVID-19 confirmed by positive PCR test for SARS-CoV-2, 2 patients had COVID-19 diagnosed by clinical symptoms and suggestive CT scan and 3 (4%) did not meet diagnostic criteria for COVID-19. The 2 COVID-19 patients with negative RT_PCR did not receive antimicrobials prior to MDR colonization. Two out of the 3 non-COVID-19 patients received antimicrobial therapy prior to MDR acquisition. None of these 5 patients acquired MDR during ICU stay.

Although we agree with the reviewer that comparing non-COVID-19 and COVID-19 patients admitted in the same ICUs would be very interesting, the low number of such patients in our cohort precludes any statistical analysis, and in our view does not allow to draw any valuable conclusion.

We precise in the result section the number of confirmed COVID-19 patients (page 8 line 179).

3.Comparison between patients admitted to COVID 19 ICUs and control admitted to medical-surgical ICUs: as available in Table 2 some variables such as age, length of antimicrobial therapy, transfer from another hospital and SAP 3 score are significantly different between cases and controls (p<0.001, not considered in multivariable competing risk analyses for factors related to MDRB acquisition). May these discrepancies influence statistical significance in MDRB acquisition between the two groups?

We thank the reviewer for his/her comment that was also raised by others reviewers. Table 2 compares the covid-19 cohort with the control cohort and we agree that there are some differences. However, all the variables that could be associated with MDR acquisition were tested in univariate analysis mixing the 2 cohorts (sup Table 1) and in the competitive risk analysis. All the variables that had a p<0.10 were included in the adjusted competitive risk analysis (Table 3). Nonetheless, as numbers are not very high, confounders may still exist. We precise it as a limit at the end of the discussion (page 19 line 383)

About large antibiotic use during viral pandemic: authors observed a large antibiotic use in their study, which was significantly associated with a higher risk MDRB acquisition in COVID-19 patients. In our experience, matured in COVID-19 Infectious Disease Unit during pandemic, antibiotics were generally prompted discontinued at the moment of patients transfer to our Unit from the Emergency Room. Currently there is a lack of data about large spectrum antimicrobial use in COVID 19 ICUs and medical wards during first wave pandemic, thus representing a limitation of this study and an area of further research.

We thank the reviewer for his/her interesting suggestion. We analyzed the spectrum of the main antimicrobial therapies administered in our cohort during the exposure time of the patients. Comparing MDR + and MDR – patients we observed that the higher risk of MDR acquisition was mainly driven by pipe-taz administration, however this was not statistically significant (chi-2 test). Moreover, this result was not adjusted to potential confounders and we fully agree with the reviewer that this is an important area for further research.

We provide the comparison of main AMT according to MDR acquisition in the table S2 of the ESM and refer it in the result section (page 8, line 194). We discussed the potential role of AMT spectrum in the discussion section (page 18 line 354).

We agree with the reviewer on the wide use of AMT in our cohort. Just note that the AMT use and duration indicated in the table gather the initial AMT but also AMT administered during ICU stay before MDRB acquisition.

In conclusion, this monocentric retrospective case control study leads the way to further research on the acquisition of multidrug-resistant bacteria in patients admitted to the wards during viral pandemics.

Reviewer 2 Report

the paper explore an interesting question related to MDR bacteria infection in COVID 19 patients in ICU.

the paper itself interesting however has some bugs that need clarification.

The authors need to clarified the use of immunosuppressive drugs in the COVID-19 patients , such as corticosteroi or drug inhibite ingflamatory cascade.

The authors need to clarified if used immunosuppressive drugs in the COVID-19 patients , such as corticosteroid or drug inhibit inflammatory cascade (tocilizumab , canakinumab etc,), because this drugs can be contribuite to MDR acquisition.

if immunosuppressants were used, the authors should analyze the infection data by separating patients according to their temporal use and according to the type of action of these drugs.

the comparison with the control group (shown in table 2) is necessary but is affected by the control population used.
the table shows a clear difference between the two groups in the use of CVC, Urinary cather, renal replacement, antimicrobiological use; the authors should explain these differences and analyze when each single factor can contribute to the infection data

the authors should clarify whether the control population is a valid population for comparison

the conclusions should therefore respect these new data

Author Response

the paper explores an interesting question related to MDR bacteria infection in COVID 19 patients in ICU.

the paper itself interesting however has some bugs that need clarification.

The authors need to clarified if used immunosuppressive drugs in the COVID-19 patients , such as corticosteroid or drug inhibit inflammatory cascade (tocilizumab , canakinumab etc,), because this drugs can be contribuite to MDR acquisition. If immunosuppressants were used, the authors should analyze the infection data by separating patients according to their temporal use and according to the type of action of these drugs.

We thank the reviewer for his/her comment that gives us the opportunity to clarify our study. In this study none of the patients received biological immunomodulators (anti IL-6, anti-IL-1 etc.)). We had a small number of patients that received corticosteroids therapy before MDRB acquisition N=7 and 3 out of these 7 acquired MDR bacteria during ICU stay.

Given this very low numbers, our study was not biased by immunosuppressive drugs. We add these information in the result section and in the Table 1. We mention in the discussion that immunosuppressive treatment may increase furthermore the risk (page 18 line 352)

the comparison with the control group (shown in table 2) is necessary but is affected by the control population used.
the table shows a clear difference between the two groups in the use of CVC, Urinary cather, renal replacement, antimicrobiological use; the authors should explain these differences and analyze when each single factor can contribute to the infection data.

the authors should clarify whether the control population is a valid population for comparison

R #1 and R#3 also raised this question. Indeed, in the Table 2 that compares the covid-19 cohort with the control cohort, we agree that there are some differences. Control patients were matched on 2 important criteria: period of exposure and mechanical ventilation. It was not feasible to match on all the factors linked to MDR acquisition.

Nonetheless, all the variables that could be associated with MDR acquisition were tested in univariate analysis mixing the 2 cohorts (sup Table 1) and in the competitive risk analysis. All the variables that had a p<0.10 were included in the competitive risk analysis (Table 3).

We agree that the control population is not strictly similar, control patients were matched on the main criteria that are associated with MDR risk. It was not feasible to match on all these factors. To address this problem, we conducted a multivariable model using the Fine and Gray method (competitive event: death) adjusting for these variables (table 3). In the supplementary electronic Table S1 we compare patient that acquired MDR bacteria from those who didn’t considering all patients (Covid-19 and SAH).

Moreover, as requested by R#3 we tried to build another control cohort of septic shock patients and observed that this would be a worse control population due to higher imbalance between the 2 cohorts (see response to R#3).

the conclusions should therefore respect these new data

Reviewer 3 Report

Proposed paper is really interesting and well conducted. I have a major comment regarding control Group (first three comments) and some minor issue that need to be answered before it can be acepted for pubblication:

  • Controls comes from a very different pathology. Every severe disease can alter the immune system and facilitate MDR infection but a Group composed of severe septic shock could be something more similar to the risk of the COVID patients. If this is not available please describe this such a limitation. In fact rate of antibiotic use and so of possible MDR infection could be different in this two groups.
  • On the same way control were recruited in a very wide time range (2016-2019) while COVID subject only in few months. It could be possible that local epidemiology of MDR changes over time and so differences could be determined also by this. Please discuss.
  • Differences in very important item are present between COVID and controls such as hospitalization in the previous 6 months, the use o devide (urine and venous cathether), renal replacement therapy, antimicrobial therapy (as described above). This differences could drive the absence of significant difference. Please discuss and try to found a more similar control Group if possible.
  • Please complete the analysis with a mutivariable model in which the possible significant determinants of being infected by a MDR bacteria were defined (vasopressor use? longer stay?). This should be done inside the COVID Group only.
  • An important paper regarding COVID-19 mortality has been missed and need to be cited, i.e. Hypertension. 2020 Aug;76(2):366-372.

Author Response

Proposed paper is really interesting and well conducted. I have a major comment regarding control Group (first three comments) and some minor issue that need to be answered before it can be accepted for publication:

  • Controls comes from a very different pathology. Every severe disease can alter the immune system and facilitate MDR infection but a Group composed of severe septic shock could be something more similar to the risk of the COVID patients. If this is not available please describe this such a limitation. In fact rate of antibiotic use and so of possible MDR infection could be different in this two groups.

We thank the reviewer for its interesting comment. We agree that the control group comes from a very different pathology. However, MDR acquisition in the ICU is mainly driven by the time of exposure, presence of invasive procedure, global infection control process and antimicrobial therapy pressure rather than the underlying pathology.

We fully agree that septic shock and COVID-19 are inflammatory processes that induce a form of ICU-acquired immunosuppression, however this also the case for subarachnoid hemorrhage. Indeed, systemic inflammation occurs in up to 88% of SAH patients (Festic, E. et al. Neurocrit Care 2014 ; 20 :375–381). Immunosuppression also occurs after SAH (Sarrafzadeh A, et al. Stroke 2011 ;42:53-80)  affecting mostly the cellular component of the immune system response (Roquilly A, et al. PLoS One 2013;8: e71639). As in COVID-19, lymphopenia is a common finding after SAH (Giede-Jeppe A et al. J Neurosurg 2019;132: 400-407 ; Al-Mufti F, et al. J Neurointerv Surg 2019;11: 1135-1140). We looked on lymphocyte count in both cohorts and observed that 45% of SAH patients, had lymphopenia in the first 5 days of admission (< 1000/mm3). The median lymphocyte count in these patients was 1049 (771-1555)/mm3.

In COVID-19 patients 48% had lymphopenia and the median lymphocyte count was 1075 (730-1743), p =0.90 calculated by the Mann-whitney test. We add these values in Table 2.

Nonetheless, taking into account reviewer comment, we tried to build a 2nd control cohort of septic shock patients. 203 septic shock patients were admitted in our ICU between 2016 and 2019 but we were able to match the duration of exposure in only 60 patients to the covid-19 cohort. The 12 non-matched patients (17%) belong to those with the longest ICU LOS (LOS range of the 12 patients 26-63 days) creating a bias in the time at risk.

Moreover, the 2 cohorts had several imbalance: i) comorbidities were more frequent in the septic shock cohort than in the Covid-19 cohort: Charlson index 5 (3-6) as compared to 1 (0-3) ; ii) in accordance, previous hospitalization was much more frequent in septic shock patients (50% vs 15%). iii), antimicrobial therapy was prescribed in 98% of the septic shock patients compared to 78% in the COVID-19 cohort and iv) MDR colonization on admission was present in 12% of the septic shock compared to 8% of the COVID-19 patients.

These data highlight our arguments that SAH patients constitute better controls to adjust for confounders aiming at evaluate the MDR acquisition risk. Indeed they have less comorbidities and have a higher LOS than septic patients. We thus keep the SAH cohort, but as we considered that the results for the septic shock patients may be interesting for some of the readers, we add the septic shock cohort description in the ESM.

Following reviewer suggestion, we analyzed the MDR acquisition in the septic shock cohort, we observed that 19 patients acquired 44 MDRB during ICU stay (32%) and that the rate of MDRB acquisition was 31/ 1000 patient-days. These rough value are very similar to those of COVID-19 patients (30/1000 patient-days) despite the fact that the septic cohort had probably a higher risk at baseline.

We modified our manuscript accordingly:

we briefly describe the septic shock control cohort in the method section

We mention their MDRB acquisition rate in the result section (page 13 line 240) and in the new figure S2

We added the description of the 60 septic shock patients with COVID-19 patients in the ESM as a Table S3

We modified the discussion section page 19 / line 360

  • On the same way control were recruited in a very wide time range (2016-2019) while COVID subject only in few months. It could be possible that local epidemiology of MDR changes over time and so differences could be determined also by this. Please discuss.

We thank the reviewer for this important comment. We analyzed the rate of MDR acquisition by tertile in the 2 cohorts to try to assess this problem.

The rates of MDR acquisition per tertile (N= 24) in the SAH cohort were:  22/1000, 25/1000 and 11/1000 patient-days.

The rates of MDR acquisition per tertile (N=20) in the septic shock cohort were: 28/1000 , 30/1000,  and 36/1000 patient-days.

Thus, with all the limitation given the limited number of patients per year, we didn’t observe a clear increase of MDR acquisition risk in our unit across the last years.

We have added these date in the Figure S2 that shows the rate of MDR acquisition over the 4-year period (2016- 2019) in both control cohorts and the COVID 19 rate as a dotted line.

We also mention briefly these results in the result section

  • Differences in very important item are present between COVID and controls such as hospitalization in the previous 6 months, the use o devide (urine and venous cathether), renal replacement therapy, antimicrobial therapy (as described above). This differences could drive the absence of significant difference. Please discuss and try to found a more similar control Group if possible.

R #1 and R#3 also raised this question. Indeed, in the Table 2 that compares the covid-19 cohort with the control cohort, we agree that there are some differences. Control patients were matched on 2 important criteria: period of exposure and mechanical ventilation. It was not feasible to match on all the factors linked to MDR acquisition.

Nonetheless, all the variables that could be associated with MDR acquisition were tested in univariate analysis mixing the 2 cohorts (sup Table 1) and in the competitive risk analysis. All the variables that had a p<0.10 were included in the competitive risk analysis (Table 3).

As suggested in the previous remark, we tried to build another control cohort with septic shock patients; these patients had nearly 100% central venous line, urinary catheter and initial antimicrobial therapy. As discuss above, although this specific population had a higher risk at baseline to acquire MDR, its acquisition rate was similar than the COVID-19 patients reinforcing in our view the message that patients admitted in ICU during a pandemic crisis have a high risk of MDRB acquisition.

  • Please complete the analysis with a mutivariable model in which the possible significant determinants of being infected by a MDR bacteria were defined (vasopressor use? longer stay?). This should be done inside the COVID Group only.

With all respect due to the reviewer, we believe it is not feasible to perform the multivariate model as suggested. Indeed, our statistician advised us, that due to the small sample size and the limited number of events he was unable to perform a reliable multivariable model in the COVID group only.

  • An important paper regarding COVID-19 mortality has been missed and need to be cited, i.e. Hypertension. 2020 Aug;76(2):366-372.

As requested, we added the reference.

Round 2

Reviewer 3 Report

Authors replies to all the query raised. Paper can now be accepted for pubblication.